# One-Carbon and Polyamine Metabolism as Cancer Therapy Targets

**DOI:** 10.3390/biom12121902

**Published:** 2022-12-19

**Authors:** Anowarul Islam, Zeeshan Shaukat, Rashid Hussain, Stephen L. Gregory

**Affiliations:** 1College of Medicine and Public Health, Flinders University, Adelaide 5042, Australia; 2Clinical and Health Sciences, University of South Australia, Adelaide 5001, Australia

**Keywords:** methionine, one-carbon metabolism, polyamines, cancer, metabolic therapy, reactive oxygen species, autophagy

## Abstract

Cancer metabolic reprogramming is essential for maintaining cancer cell survival and rapid replication. A common target of this metabolic reprogramming is one-carbon metabolism which is notable for its function in DNA synthesis, protein and DNA methylation, and antioxidant production. Polyamines are a key output of one-carbon metabolism with widespread effects on gene expression and signaling. As a result of these functions, one-carbon and polyamine metabolism have recently drawn a lot of interest for their part in cancer malignancy. Therapeutic inhibitors that target one-carbon and polyamine metabolism have thus been trialed as anticancer medications. The significance and future possibilities of one-carbon and polyamine metabolism as a target in cancer therapy are discussed in this review.

## 1. Introduction

For cancer cells to multiply quickly and unchecked, altered metabolism is essential, and a range of modifications to their metabolism are known to enable increased survival and multiplication. A key element in this is the generation of sufficient nucleotides and lipids, both of which are dependent on the availability of methyl groups from the one-carbon metabolic pathways. These methyl groups are necessary for the biosynthesis of compounds such as nucleic acids, amino acids, and the major membrane lipid phosphatidyl choline [1,2], among many others.

The methionine and the folate cycles (Figure 1) are crucial interrelated pathways in one-carbon metabolism that provide methyl groups for the creation of DNA, amino acids, creatine, polyamines, and phospholipids [3]. Nucleotide metabolism and epigenetic regulation of DNA and histones, whose aberrant expression is a distinguishing feature of tumor cells, both depend on one-carbon metabolism to maintain genomic integrity. Studying one-carbon metabolism offers the prospect of precision medicine intervention for disease prevention, the discovery of biomarkers, and the diagnosis and treatment of different illnesses, particularly cancer [1,4].

## 2. One-Carbon and Polyamine Metabolism

### 2.1. The Methionine Cycle

The first phase of the methionine cycle is the synthesis of S-adenosylmethionine (SAM) from methionine using the enzyme methionine adenosyl transferase (MAT) [5,6]. While MAT II (a dimer) is expressed in the majority of other cell types and is encoded by MAT2A, MAT I (a tetramer) and MAT III (a dimer) are often expressed in the liver, where substantial SAM synthesis occurs [7,8]. Then, SAM is used by numerous methyl transferases to donate a methyl group to their diverse targets. This loss of a methyl group changes SAM into S-adenosylhomocysteine (SAH). To complete the methionine cycle, SAH is hydrolyzed to homocysteine by the enzyme SAH hydrolase (AHCY or SAHH) [5,6]. Homocysteine can then be re-methylated to methionine by the enzymes methionine synthase (5-methyltetrahydrofolate-homocysteine methyltransferase; MTR or MS) or betaine-homocysteine methyltransferase (BHMT). Alternatively, cystathionine-β-synthase (CBS) can divert homocysteine into the transsulfuration route to become cystathionine, which is subsequently changed into cysteine by cystathionase (CTH) for use in the synthesis of glutathione and the preservation of redox equilibrium [5,9]. To summarize, SAM is primarily used to donate a methyl group, then is either recycled to methionine by receiving a methyl group from the folate cycle or is converted to cysteine/glutathione. Methyl groups, a single carbon plus three hydrogens, are usually stable and unreactive, so the use of SAM and methyl transferases is essential for a wide range of biosyntheses and modifications that regulate gene expression, epigenetics, detoxification, and more [10]. To maintain metabolite levels, the methionine and folate cycles are closely connected [11]. SAM inhibits the enzymes methylenetetrahydrofolate reductase (MTHFR) and (betaine-homocysteine methyltransferase) BHMT to limit the conversion of homocysteine to methionine, allowing homocysteine to be diverted for transsulfuration when SAM is abundant, a sign of high methionine levels [8,11]. Additionally, SAM stimulates CBS, which directs homocysteine into transsulfuration [11,12]. Low methionine levels cause SAM levels to drop, freeing inhibition of MTHFR and BHMT and restraining activation of CBS to keep the methionine cycle in flux and regenerate SAM. 5-methyltetrahydrofolate (5-mTHF) builds up as a result of low methionine synthesis, and it inhibits glycine N-methyltransferase (GNMT), which would otherwise be a significant sink for SAM [11]. These feedback inhibitions act to maintain homeostasis in SAM levels.

### 2.2. The Folate Cycle

The water-soluble B vitamin folic acid is obtained from food and transformed by the body into tetrahydrofolate (THF). Through folate-mediated one-carbon metabolism (FOCM), THF can provide the necessary nucleotides for replication and one-carbon groups for DNA methylation, which are important for epigenetic gene regulation [13]. Serine is a key methyl donor in the folate cycle, though there are many other ways that cells can obtain one-carbon groups, including choline, betaine, glycine, histidine, and sarcosine [14,15]. THF can either be used for nucleotide synthesis or regenerate methionine from homocysteine in the one-carbon cycle (Figure 1). So, it can be seen that FOCM regulates the production of S-adenosylmethionine (SAM), nucleotides, certain amino acids, glutathione, and other cellular processes critical for the proliferation of cancer cells [15]. FOCM distributes carbon atoms among the various acceptor molecules required for biosynthesis in addition to controlling the nutritional status of cells through their redox and epigenetic states.

### 2.3. Polyamine Synthesis

The other major metabolic pathway that relies on SAM is the synthesis of polyamines. Spermidine, putrescine, and spermine are polycationic alkylamines that interact with negatively charged macromolecules [16] because they have protonated amino groups at physiological pH levels. They are involved in a number of cellular processes, such as chromatin organization, cellular proliferation, gene regulation and proliferation, immune system function, and cell death [17,18,19,20]. All cells produce polyamines in their cytoplasm, and their synthesis requires SAM plus ornithine, an amino acid from the urea cycle [21]. SAM is decarboxylated by SAM decarboxylase (SAMDC) to generate s-adenosyl methioninamine or dcSAM, which is a key aminopropyl donor used to form spermidine (Figure 1). The other part of spermidine comes from ornithine via ornithine decarboxylase (ODC), which generates putrescine. Putrescine plus dcSAM is used by spermidine synthase to generate spermidine. A further aminopropyl group from dcSAM can be added to spermidine by spermine synthase to generate spermine, the final product in this pathway. We do not address the interesting topic of polyamine degradation in this review; it is covered in detail elsewhere [16,17]. To this point, we have considered the three main outputs of one-carbon metabolism: methyl groups, cysteine/glutathione, and polyamines. We now move to examine how these pathways impact carcinogenesis.

## 3. The Implications of One-Carbon and Polyamine Metabolism for Cancer

### 3.1. Folate Metabolism and Cancer

Due to its range of roles in protein and DNA synthesis, methylation processes, and redox homeostasis, folate metabolism can contribute to oncogenesis. In tumor treatment, drugs that specifically target folate metabolism have been employed frequently, particularly against dihydrofolate reductase (DHFR) [22]. These inhibitors stop the growth of cancer by preventing the production of nucleic acids, which are needed for DNA replication and cell proliferation. DHFR inhibitors block the production of tetrahydrofolate, which thus inhibits purine and thymidylic acid synthesis [22]. However, antifolate medications have an adverse effect on normal cells when used to treat cancer because one-carbon metabolism is also required for healthy cells, particularly in the immune system. Nonetheless, numerous cancers have been treated with DHFR inhibitors, such as methotrexate, which was introduced in 1947 but is still very widely prescribed. Like other chemotherapeutic treatments, these drugs may fail because cells develop resistance by, for instance, impairing drug absorption, decreasing drug retention inside the cell, and decreasing drug affinity [23]. There is a need to develop further therapies that specifically target folate metabolism.

In a review of the mRNA profiles of 1981 tumors, MTHFD2 and SHMT2 were shown to be among the top five genes with the highest levels of expression, demonstrating the carcinogenic influence of mitochondrial folate metabolism [24,25]. Similar studies on the mitochondrial folate metabolism enzymes revealed a link between cancer and aberrant SHMT2 and MTHFD2 expression [26,27]. Aberrant SHMT2 and MTHFD2 expression might impair DNA synthesis and damage redox balance, which is important for cancer cell survival [28]. Other folate metabolism enzymes, such as SHMT1 and MTHFD1L, have also reportedly been linked to cancer. Disrupting SHMT1 interferes with the incorporation of dUMP into DNA, causing DNA double-strand stability to be disturbed [29]. Additionally, ovarian cancer is prevented from spreading and growing by SHMT1 knockdown [29]. Lung cancer cells are also affected by SHMT1 knockdown [30]. According to a recent study, MTHFD1L knockdown caused tongue squamous cell carcinoma cells to die under redox stress via lowering the concentration of NADPH [31]. These results suggest that folate metabolism is a desirable target for the therapy of cancer if the problems of toxicity and resistance can be overcome.

### 3.2. Serine Metabolism in Cancer

Changes in serine metabolism may have significant consequences that may lead to the development of cancer as well as other illnesses [32,33]. Serine can be absorbed by the cell or produced by the serine synthesis pathway from glycolytic intermediates. It has long been recognized that serine, whether from diet or generated endogenously, is linked to cancer, and actively promotes its growth [34,35]. Serine can also be produced by breaking down cell proteins, such as through autophagy, and by converting glycine [36]. The process of serine synthesis (SSP) is one of numerous glycolysis side branches that allows carbons obtained from glucose (or pyruvate under gluconeogenic circumstances) to be redirected to the production of serine and is upregulated in many cancers [37]. Glucose is the primary source of carbons for de novo serine synthesis in people and rats that are well-fed, but under starving conditions, gluconeogenesis can contribute up to 70% of the total serine produced [38].

Serine is necessary for the creation of phospholipids such as sphingolipids and phosphatidylserine, as well as other amino acids like cysteine and glycine. Serine is a key methyl donor, though there are many other ways that cells can obtain one-carbon groups, including choline, betaine, glycine, histidine, sarcosine, and the formate that is produced when tryptophan is broken down [14,15]. Studies in yeast and mammalian cells revealed that serine catabolized in the mitochondria is the source of the majority of the cytosolic one-carbon units [29,39,40], and blocking one-carbon metabolism in both the mitochondria and cytoplasm precludes cell growth [37].

Serine’s role in generating methylene-THF makes it a key contributor to avoiding the toxic consequences of homocysteine build-up. Homocysteine is the link between the transsulfuration pathway and the methionine cycle, and the building blocks for the synthesis of cysteine are homocysteine and serine. Serine depletion results in lower amounts of glutathione [41] because glycine and cysteine are by-products of serine degradation, whereas activation of serine synthesis enables glucose-derived carbon to be channeled towards glutathione synthesis for antioxidant defense [32,42]. This has implications for tumor oxidative stress tolerance that have not been fully examined (see 4.2 below).

### 3.3. SAM-S Metabolism in Cancer

Methionine, which makes up half of the body’s daily requirement for amino acids, is the primary amino acid used in the liver to produce SAM [43,44]. SAM is produced by MAT (SAM synthase) from methionine in an ATP-dependent mechanism [43]. The adenosyl moiety of ATP is combined with methionine during this process to change it into a high-energy reagent that can carry a sulphonium ion. SAM can then transfer a methyl group to a variety of substrates, including proteins, DNA, RNA, and lipids [45]. The cellular level of SAM can be affected by impaired dietary intake, absorption, transport, metabolism, or enzymatic processing of methionine [6,46,47,48]. For instance, dietary methionine limitation lowers SAM levels and increases the longevity of certain species [49,50,51].

Because cancer is frequently characterized by abnormal methylation states and methionine or SAM dependency, SAM has been explored as a therapeutic target in the treatment of cancer [52,53]. For example, rats have been used in tests to determine how SAM treatment affected the growth of neoplastic liver lesions. The percentage of the liver that was occupied by GST-P-positive lesions significantly decreased when SAM was administered to rats during the clonal expansion of initiated cells (promotion), primarily as a result of a reduction in the size of the lesions [54,55,56,57,58,59,60]. The number and size of liver nodules decreased after receiving the same SAM doses for 11 weeks [54,55]. A consistent decrease in incidence and multiplicity of neoplastic nodules could be observed when SAM medication was continued for up to six months [61]. On a cellular level, SAM’s chemopreventive action is linked to an increase in remodeling and a dose-related reduction in DNA synthesis in preneoplastic and neoplastic lesions [54,58,59]. Additionally, rats given SAM showed an increase in apoptosis in neoplastic nodules and hepatocellular carcinoma [55,58]. SAM therapy decreased carcinogenesis and metastasis in vivo while increasing apoptosis and decreasing the proliferation and invasiveness of breast cancer cells in vitro [62]. SAM treatment has been shown to be effective in inhibiting the proliferative and invasive potential of many cancer cell lines [63,64]. SAM selectively inhibits the proliferation and invasiveness of liver cancer cells by changing the transcriptome and methylome [65]. Although SAM has positive impacts on the treatment of cancer, more research is needed to establish SAM as a cancer therapy, as in many cases, the specific metabolic changes responsible for the observed anti-cancer effects are unclear.

### 3.4. Methionine Dependency in Cancer

Methionine metabolism and cancer have been linked on several levels. Even though they easily convert homocysteine into methionine, the majority of cancer cells are unable to proliferate if methionine in the media is replaced by homocysteine. Surprisingly, intracellular methionine levels in breast cancer cells remained substantially stable when they were transferred to homocysteine media and analyzed; however, in this situation, SAM levels were strongly depleted [66]. Homocysteine substitution for methionine has no effect on non-cancerous cells, suggesting they have less need for SAM. Cancer and normal cells are different in their growth rates with different metabolic needs, so it is frequently challenging to interpret the differences between the metabolic dependencies of normal and cancer cells. Perhaps unsurprisingly, there are some methionine-independent tumor cell lines, and in these cases, SAM levels are relatively normal [67,68].

According to the Hoffman effect, methionine is metabolized differently by cancerous and non-tumorigenic cells. Using ^11^C-methionine positron emission tomography, human cancers may be easily seen and distinguished from normal tissue, demonstrating this higher need (Met-PET). Met-PET imaging often outperforms 18F-deoxyglucose PET (FDG-PET) imaging, particularly in glioma, as the increased brain glucose metabolism interferes with tumor-specific FDG signals. However, multiple myeloma and other malignancies have also been studied with Met-PET [69].

### 3.5. Homocysteine Metabolism and Cancer

Hyperhomocystinuria and cancer have been shown to be closely related by recent scientific developments. Homocystinuria is defined by a rise in the level of homocysteine (Hcy) in the serum and can come from an inborn mistake in the metabolic pathways of sulfur-containing amino acids [70]. Cancer patients have also been found to have increased plasma homocysteine concentrations. There are strong clinical correlations between a number of polymorphisms in the enzymes implicated in the Hcy detoxifying pathways and various cancer types [71,72,73,74,75,76,77,78,79,80,81]. Many cancer types exhibit high plasma Hcy levels in the advanced stages, although there may be little to no change in plasma Hcy levels in the earlier stages of the disease [73,82,83,84,85,86,87,88,89,90,91]. Why the Hcy levels differ between the early and late stages of cancer is unclear. However, since Hcy promotes the growth of cancer cells [92], increased generation and secretion of Hcy seems likely to be an adaptive metabolic mutation acquired during cancer progression. Caco-2 cell lines with higher homocysteine levels exhibit greater cellular proliferation. By including more folate in the culture media or by supplementing it with its metabolites, such as 5-MTHF [93], this increased proliferation can be reduced. However, because a very high Hcy concentration may potentially be lethal to the cancer cells, advanced-stage cancer cells may release Hcy. Clinically, the situation is less clear—in some studies, there is no evidence of a correlation between Hcy levels and cancer risk [94]. Further investigations are required to reveal the precise mechanism of how cancer patients deal with Hcy metabolism.

### 3.6. The Role of One-Carbon Metabolism in Nucleotide Synthesis in Cancer

The synthesis of purine and pyrimidine nucleotides, which are required for the synthesis of DNA and RNA, depends on the one-carbon cycle [95]. A single carbon, typically from serine, is transferred during one-carbon metabolism to create the pyrimidine and purine bases [52], hence the significance of serine in the production of nucleotides. During glycolysis, serine is produced from 3-phosphoglycerate (3-PG) [96]. Serine-derived one-carbon transfer to tetrahydrofolate results in 5,10-methylenetetrahydrofolate (CH2-THF), a substance essential for the synthesis of pyrimidines [97]. CH2-THF is also the methyl donor used to regenerate methionine from homocysteine, so there is a balance between its use in pyrimidine synthesis versus providing the methyl group to SAM for use in DNA or protein methylation, polyamine synthesis, or the generation of glutathione.

The subsequent transformation of CH2-THF into 10-formyltetrahydrofolate (CHO-THF) is an essential component of purine synthesis [97]. Therefore, the synthesis of both pyrimidines and purines depends on a carbon donor such as serine and the tetrahydrofolate carrier. Due to the need for a large quantity of DNA nucleotides, one-carbon metabolism is crucial for cancer cells to proliferate quickly. As a result, one-carbon metabolism is a prospective target for reducing cell growth. It was shown that lowering serine levels or blocking particular mitochondrial folate metabolic enzymes decreased the number of purine nucleotides, which in turn prevented proliferation [41,98,99]. As a result, researchers are actively looking at anticancer medications that target one-carbon metabolism [100,101].

### 3.7. Polyamine Metabolism in Cancer

Prostate cancer cell proliferation and differentiation, often controlled by androgen hormones, are correlated with levels of polyamines, particularly spermine [102] which is plentiful in the human prostate. Spermine may serve as a biomarker to distinguish between low-grade and high-grade prostate cancers because its content is lower in the latter [103]. In prostate cancer, the most significant metabolic disturbance observed was increases in polyamine metabolites and in dcSAM [104]. The PTEN-PI3K-mTOR complex 1 (mTORC1) pathway was shown to regulate the stability of SAMDC (AMD1), which controls the use of SAM for polyamine synthesis in prostate cancer [104,105]. Inhibitors of mTORC1 or SAMDC were able to significantly impede growth in prostate cancer cell lines, and this could be partly rescued by supplementing with spermidine. In this case, the role of ODC1 in polyamine regulation downstream of mTORC signaling was excluded—it was just via SAMDC regulation. However, in c-MYC transgenic mice, c-MYC has been shown to promote prostate cancer carcinogenesis by boosting polyamine production through the transcriptional control of ODC [106]. This is significant because ODC1 has been identified as a c-Myc-responsive rate-limiting step in polyamine synthesis [107]. Notably, PGC-1α inhibits c-MYC and hence ODC, which reduces polyamine production and lowers the aggressiveness and spread of prostate cancer [106]. By contrast, the androgen receptor typically acts in prostate cancer to upregulate ODC1 expression [108], and indeed ODC1 overexpression alone may be enough to drive prostate tumorigenesis [109].

Similar to the observation in prostate cancer, human breast cancer tissue has a lot more acetylated polyamines than healthy breast tissue [110]. In breast cancer patients, estrogen signaling is linked to the creation of polyamines and purines. Estrogen directly contributes to the progression of breast cancer by activating the estrogen receptor (ERα), which binds to estradiol (E2) [111]. Through the mitochondrial folate route, this binding activates ERα and causes the activation of genes that boost polyamine and purine production [111,112]. Additionally, due to their effects on the activity of the insulin receptor, polyamines may control the mitogenic action of insulin in breast cancer [113]. ODC mRNA and protein levels are markedly increased in breast cancer patients, and they positively correlate with the tumor, node, and metastases (TNM) stages of the disease. Increased ODC activity is linked to higher cancer cell proliferation rates and a worse prognosis for breast cancer patients [114]. Arginase, which changes arginine into ornithine [115,116], is more prevalent in breast cancer, making it a potential market for breast cancer in its latter stages [117]. In addition to ODC, breast cancer also exhibits increased levels of ADC and agmatinase, enzymes involved in the synthesis of putrescine from arginine [118]. Early in metastasis, arginase and polyamine production are increased [119]. These considerations are relevant to this review because, in each of these cases in which polyamines are elevated in cancer, SAM and the one-carbon cycle are required for their synthesis.

Patients with pancreatic cancer have polyamines found in their urine, serum, and saliva, which makes them potential biomarkers [120,121,122,123]. In human pancreatic ductal adenocarcinoma (PDAC), KRAS mutations are the most prevalent (representing around 95% of all mutations) [124]. In addition, the copy number of c-MYC has increased in more than 50% of human PDAC cell lines [125]. Similar to other cancers, KRAS and MYC are upstream activators of polyamine production in PDAC [124,126]. ODC levels rise in pancreatic cancer and aid in the development and spread of advanced pancreatic cancer [127,128,129]. Employing an ODC inhibitor (DMFO) and a polyamine transport inhibitor (Trimer44NMe) together greatly decreased the survival of PDAC cells by inducing apoptosis [126]. Immune privilege must be established in order for the PDAC tumor to survive, and spermine is critical for this process [130].

Poor prognosis is linked to the dysregulation of polyamines in neuroblastoma, and various polyamine homeostasis-related genes are transcriptional targets of cMYC/MYCN [131,132,133]. The modulation of the SLC3A2 polyamine exporter and other essential elements of the polyamine pathway in vitro is directly induced by MYCN, leading to increased polyamine production and accelerated neuroblastoma cell proliferation [134]. ODC has been recognized as a potent oncogenic transforming factor, and in neuroblastoma, it is the most well-studied target of the transcription factor c-MYC/MYCN [133,135,136]. In vivo neuroblastoma cell proliferation and MYCN-mediated oncogenesis are both reduced in animal models when ODC is disabled [137]. Along with ODC, SAMDC is a target of MYCN and plays a significant role in the growth of neuroblastomas [138,139]. In murine neuroblastoma, S-adenosylmethionine synthetase overexpression is linked to the development of treatment resistance [138]. Transgenic mice used in a preclinical study that used polyamine antagonist regimens targeting ODC1 and SAMDC had their neuroblastoma initiation reduced [140,141].

Metabolic enzymes and polyamine levels affect both treatment and prognosis in leukemia [142]. High levels of polyamines are linked to a bad prognosis in leukemia cells. However, polyamine depletion in healthy cells also results in cell cycle arrest, highlighting the need to preserve polyamine homeostasis. Patients with acute lymphoblastic leukemia (ALL) have increased ODC activity and putrescine levels, and their recurrence can be detected by increased spermine levels in erythrocytes [142].

Polyamine depletion is a plausible approach to decreasing polyamine levels in cancer. Overexpression of the polyamine acetyltransferase SSAT drives the first step of polyamine breakdown and can result in diminished cell growth, migration, and invasion by blocking AKT and GSK3b signaling [143]. These findings were made using a variety of colon carcinoma cell models and human hepatocellular malignancies.

It is not new to use polyamines and their metabolites as cancer biomarkers [144]. In lung and liver malignancies, polyamines and their metabolites in the urine and plasma can be helpful both for diagnosis and as indicators of tumor development [145,146]. Diacetylspermine has been linked to lung and ovarian cancers as a reliable urine biomarker [147,148,149]. Right-side colon tumors associated with biofilms have also been shown to contain significant quantities of diacetylspermine [150,151]. Urinary or serum measurements of polyamines and polyamine metabolites have demonstrated potential as biomarkers for colon, pancreatic, and prostate malignancies [120,152,153,154,155]. The development of more individualized methods for cancer diagnosis and therapy based on the use of polyamines as biomarkers may be aided by such analyses in conjunction with increasingly accurate genetic signatures.

## 4. Mechanisms Relating One-Carbon and Polyamine Metabolism to Cancer

### 4.1. The Function of One-Carbon Metabolism in Methylation Reactions

SAM is a common methyl donor used in the methylation of RNA, DNA, and histones [65]. The methyl group typically comes from serine via CH2-THF and is then transferred to methionine, then SAM before transfer to the final target molecule [156]. DNA methylation primarily takes place at the 5’ carbon of the pyrimidine base cytosine (5 mC) in CpG islands. DNA methyltransferases (DNMTs) like DNMT3a, DNMT1, and DNMT3b catalyze DNA methylation using SAM as the methyl donor [157]. Numerous tumor cells, including colon, cervical, and breast cancer cells, have been found to exhibit hypermethylation in the DNA [158]. Reduced gene expression of tumor suppressor genes is caused by the hyper-methylation of their promoters. Additionally, it has been noted that DNA hypermethylation and chemoresistance are associated [159]. A number of clinical kits are already being produced for detecting DNA methylation in cancer patients [160,161,162,163,164,165,166,167,168], demonstrating how this correlation has been incorporated into clinical practice.

RNA methylation also occurs, primarily taking place at the N6 position of an adenine base (m6A) near a stop codon [169,170]. RNA methyltransferases like METTL3, METTL14, and WTAP catalyze the methylation of RNA using a SAM donor [171]. N6-Methyl Adenosine (m6A) in RNA has a variety of roles in the development and spread of cancer. By encouraging the translation of these mRNAs, METTL3 activity boosts and augments MYC, BCL2, and PTEN in human acute myeloid leukemia (AML) [172]. Similar findings suggest that RNA methylation fosters the development of tumors in other cancer types, including pancreatic, colorectal, hepatic, and breast cancer [173,174,175,176]. In addition, it has been noted that RNA methylation is a reliable diagnostic indicator for gastrointestinal malignancies [177]. However, RNA methylation can equally serve to increase the translation of tumor suppressors, and in these cases, overexpression of RNA methylation machinery is protective [178]. RNA methylation has also been linked to tumor immunity, so clearly, there is more work to be done to understand the full implications of RNA methylation in cancer.

In cancer cells, histone methylation and demethylation are both crucial processes. Histone methylation has received a lot of attention as a protein modification, particularly for its function in regulating gene expression. Increased methylation of H3K4, H3K36, and H3K79 frequently promotes transcription, while increased methylation of H3K9, H3K20, and H3K27 typically represses transcription [179]. AKT1, MYC, and MAPK are just a few of the cancer-related genes that are impacted by H3K4 methylation [5]. Additionally, aberrant histone methylation and altered gene expression may be caused by mutations in the histone methyltransferases MLL2, EZH2, and UTX [180,181]. In addition, cancer stem cells (CSCs) in a variety of cancer types benefit from histone demethylation via the LSD1 or Jumonji C domain families [182,183,184]. SAM depletion alters the kinetics and development of histone methylation in vivo as well as in stem cells and cancer cells [5,46,185,186,187], but it is not yet clear whether this represents a viable therapeutic opportunity.

### 4.2. Oxidative Stress and One-Carbon Metabolism in Cancer

Reactive oxygen species (ROS) levels affect the development of cancer: initiating or promoting carcinogenesis at lower levels or at higher levels leading to cell death [188]. Tumor cells typically generate relatively high levels of ROS by their aberrant metabolism and tolerate oxidative stress through several adaptations, including the generation of antioxidants such as glutathione. Glutathione can be regenerated following oxidative stress by glutathione reductase, but it requires NADPH. NADPH is generated in a number of ways, such as by activating AMPK, the Pentose phosphate pathway from glycolysis, and reductive glutamine and folate metabolism [188]. Redox-sensitive pathways are maintained in normal working order in physiological circumstances by a harmony between the creation and removal of reactive oxygen species (ROS). Oxidative stress can cause abnormal cell death and/or disease development when redox equilibrium is disrupted [189].

By restoring the activity of antioxidant defense enzymes like superoxide dismutase (SOD) and catalase and by raising levels of the anti-oxidant glutathione, cofactors of one-carbon metabolism, in particular folate and B12, have been shown to be useful in lowering oxidative damage [190]. At least in rats, a long-term reduction in the intake of folate alters the activity of Mn-SOD, catalase, and glutathione peroxidase, as well as causing irreversible oxidative DNA damage [191]. Conversely, adding dietary folate may protect against oxidative stress [192,193]. The mechanisms have not always been identified in these cases, but SAM is known to boost SOD and glutathione-S-transferase (GST) activity and replenish glutathione levels [194], so it seems likely that a significant role of folate is to allow effective regeneration of SAM and hence glutathione when under oxidative stress. A potent antioxidant molecule, GSH is a tripeptide made of glycine, glutamate, and cysteine [195]. Cysteine catabolism via the trans-sulphuration pathway raises glutathione levels and speeds up the process of ROS detoxification [42].

Lack of dietary folate, and hence lack of methylene-THF, leads to hyperhomocysteinemia, as there is no methyl donor to use up homocysteine and regenerate methionine and SAM. Perhaps surprisingly, elevated homocysteine is not associated with elevated glutathione levels but rather with ER stress and DNA damage [196], as well as atherosclerosis and dementia. Homocysteine has some direct detrimental effects, including upregulating superoxide production by NADPH oxidase, leading to increased redox stress [197]. These deleterious outcomes underline the importance of one-carbon homeostasis, as folate is required to maintain SAM levels as well as to prevent elevated homocysteine [198].

One-carbon metabolism has come to be recognized as a significant cellular regulator of NADPH levels through the activity of MTHFD, which uses methylene-THF to make NADPH in the first step toward purine synthesis [28]. Cellular NADPH/NADP+ was lowered, and oxidative stress sensitivity was raised when either the mitochondrial or cytosolic MTHFD enzymes were depleted. In response to oxidative stress, Nrf2 activity promotes serine transit through the folate cycle, so cells produce more NADPH and the reducing equivalents required to detoxify ROS [42]. Methylene-THF is thus used both to produce glutathione via the methionine cycle as well as to maintain antioxidants in their reduced state by generating NADPH [199].

In summary, it is established that there is a strong correlation between antioxidant defense mechanisms and one-carbon metabolism. It has also been established that one-carbon metabolism has an impact on cancer progression. Consequently, it seems likely that at least one of the mechanisms by which one-carbon metabolism affects cancer outcomes will be its role in maintaining antioxidant defenses. The other principal defense against oxidative damage is the recycling of damaged molecules by autophagy, and this is also regulated by one-carbon metabolism.

### 4.3. The Linkage of Autophagy to the One-Carbon and Polyamine Metabolism in Cancer

Autophagy is induced in response to various stresses to maintain metabolic homeostasis and prevent the build-up of unnecessary or damaged cellular components [200,201,202]. The aberrant regulation of autophagy is linked to many diseases, especially in neurodegenerative disease and cancer [201,203,204], as well as in cells in which aneuploidy has been induced [205,206,207,208].

Autophagy can function as a pro-survival protective pathway in cancer cells to tolerate the effects of their increased metabolic demands for rapid cell proliferation and to respond to cellular stresses that may include genomic instability and metabolic stress [209,210,211,212]. Reduced autophagy may promote tumorigenesis by increasing DNA damage rates. Autophagy is thought to be mainly regulated by Target of Rapamycin Complex 1 (TORC1) in a nutrient-sensitive condition [213]. There are now ongoing clinical trials evaluating the combination of different modulators of autophagy with other chemotherapeutics [214,215].

Studies have shown that one-carbon metabolism is involved in the regulation of autophagy and antioxidant levels. S-adenosylmethionine (SAM) functions as a conserved metabolic switch that regulates autophagy by controlling methylation [187,216,217], sulphuration [218,219,220], and synthesis of polyamines [221,222]. Furthermore, SAM also controls the availability of natural antioxidant GSH and other sulfur-containing metabolites like cysteine [223]. GSH and cysteine are essential to reduce cancer-related oxidative damage [224]. GSH depletion and increased cellular oxidative stress can trigger the autophagic response [219,225,226].

Increased methionine levels in yeast result in the inhibition of starvation-induced autophagy through increased SAM levels and methylation of PP2A. Methylated PP2A dephosphorylates the negative regulators (Npr2, Npr3, and Iml1) of TORC1 [227]. In mammals, increased SAM levels enhanced its binding to SAMTOR, which disrupts the inhibitory complex (SAMTOR-GATOR1) of mTORC1 [227,228]. SAMTOR acts as a nutrient sensor via SAM; it links one-carbon metabolism to cellular growth and autophagy via mTORC1.

Spermidine has also been demonstrated to trigger autophagy in flies, yeast, worms, and mammalian cells [221,229]. Spermidine controls autophagy by altering the expression of the autophagy-related gene (Atg) via controlling the expression of the transcription factor elF5A and TFEB [230,231]. Spermidine also suppresses acetylation by regulating the expression of acetyltransferase E1A-associated protein p300 (EP300), which promotes the deacetylation of autophagy-related proteins [232]. In addition, spermidine also reduces the availability of acetyl-CoA, which decreases acetylation and promotes autophagy [229].

Cancer cells have altered metabolism to meet the high demands for energy which results in increased cellular stress and damage. Therefore, cancer cells have a higher dependency on autophagy and other repair mechanisms compared with normal cells. Maintaining cellular levels of autophagy prevents healthy cells from tumorigenesis by limiting tissue damage, inflammation, and genome instability, but cancer cells also utilize autophagy for tumor progression and drug resistance [233,234,235,236]. Therefore, inhibiting autophagy in cancer cells is a potential target, and clinical trials are ongoing on autophagy modulators to treat cancer, though clearly, more work needs to be done in this area.

## 5. Metabolic Cancer Therapy

### 5.1. Metabolic Therapy Targeting One-Carbon and Folate Metabolism

The relevance of FOCM has been unequivocally established, and clinics have been using related medications for many years. Numerous cancers have been treated with dihydrofolate reductase (DHFR) and thymidylate synthase (TYMS) inhibitors [22,237,238], such as methotrexate and pemetrexed. Similar to other chemotherapeutic treatments, these drugs are not ideal because cells develop resistance by, for instance, impairing drug absorption, decreasing the drug’s retention inside the cell, and decreasing drug affinity. There is a need to develop further therapies that specifically target FOCM.

Since many cancer cells appear to be somewhat dependent on the presence of exogenous serine, limiting the supply of serine may have medicinal advantages. Depletion of exogenous serine will obviously have less of an impact on tumors with increased serine synthesis enzymes, but p53 loss may increase their dependency. More than half of all malignancies have p53 mutations [239], which could lead to a tumor-specific dependency on serine availability. It is a well-known therapeutic technique to reduce phenylalanine intake in individuals with phenylketonuria [240], and it would appear that a similar strategy could be used to eliminate serine from a cancer patient’s diet. Serum levels of serine and glycine can be selectively reduced by 50% in animals fed a diet missing serine and glycine [41], in mouse studies, despite the fact that serine synthesis by organs such as the liver and kidneys [241] might have been expected to maintain circulating serine levels. Mice fed on this diet showed delayed tumor formation in xenograft experiments [41]. Combining a serine-free diet with oxidative phosphorylation inhibitors, such as the biguanides metformin and phenformin, which are used to treat type 2 diabetes, enhanced the therapeutic effectiveness of this method for treating cancer in an allograft mouse model [242]. According to experimental findings, the switch to de novo serine synthesis is followed by an increase in ROS levels. This raises the prospect that suppressing antioxidant defenses or encouraging the production of more ROS could work in conjunction with serine restriction to kill tumor cells.

Limiting de novo serine synthesis is an alternative strategy for therapeutically addressing serine metabolism, particularly in tumors that exhibit serine synthesis enzyme amplification. For instance, the availability of PHGDH inhibitors [243,244] that block serine synthesis allows for preclinical and clinical examinations in patients chosen for having tumors with amplified PHGDH. However, a study employing xenograft mice models demonstrated that PHGDH depletion alone could not suppress tumor growth, casting doubt on the efficacy of this method for treating existing tumors [245]. Another problem with this strategy will be any negative consequences that may result from preventing de novo serine production. Exploiting serine metabolism clinically for the treatment of cancer is still in its infancy. A more specific method, or combination of approaches, is expected to emerge as we gain a deeper comprehension of the regulation and activity of these pathways. However, several approaches are currently in the initial phases of preclinical examination. Therefore, we are hopeful that this area of metabolism may lead to novel therapeutic possibilities.

Dietary methionine restriction considerably slows down the growth of tumors in a number of preclinical models, including both solid tumors and blood malignancies [246,247,248,249,250]. Overall, Yoshida sarcoma survival improved as a result of their reaction to a methionine-restricted diet. Regular diet mice all died by day 12, whereas Yoshida tumor-bearing mice all lived for 30 days, with the last one passing away on day 38. These tumor-bearing mice’s body weights were unaffected by the methionine-off diet [251]. Although the results of clinical investigations utilizing diets low in methionine have been inconsistent, the endpoint data were primarily concerned with the effectiveness of plasma methionine reduction [252]. The amount of plasma methionine decreased by about 50%, and patients shed an average of 0.5 kg per week. When tumors were studied after surgery, the combination of 5-fluorouracil and methionine limitation in preoperative high-stage stomach cancer patients had a remarkable impact on tumor pathology [253]. A recombinant enzyme that breaks down methionine has been created [254,255]. The gene, methioninase (METase), was obtained from Pseudomonas putida and encoded an L-methionine-deamino-mercaptomethane-lyase. Both patient-derived xenograft (PDX) and cell-based models of several malignancies demonstrated the efficacy of METase injection [250,256,257,258,259,260]. The most promising route to practical use involves methionine restriction along with chemotherapy or radiation.

### 5.2. Therapy Targeting Polyamine Metabolism

Targeting polyamine metabolism, which is dysregulated in several types of malignancies, has been the focus of therapeutic treatments for some time. In the 1960s, methylglyoxal bis(guanylhydrazone) (MGBG) was utilized as an anticancer medication, for example, against leukemia [261,262], but its effectiveness was severely hindered by its toxicities. Later research revealed MGBG to be a SAMDC inhibitor [263], suggesting SAMDC as a possible therapy target. This effort resulted in the creation of many SAMDC inhibitors, such as 4-amidinoindan-1-one 2′-amidinohydrazone (SAM486A). As an analog of spermidine and a competitive SAMDC inhibitor, MGBG lowers spermidine and spermine levels and raises putrescine levels [262]. MGBG inhibits the development of cancer cells by triggering the mitochondrial apoptosis cascade [264]. Even though these substances exhibited antitumor activity, they were nonetheless extremely hazardous. These analogs’ antitumor activity and/or toxicity were caused by off-target effects such as antimitochondrial activities in addition to interference with polyamine metabolism. SAMDC can be rendered inactive by more potent inhibitors, such as 5′-(((Z)-4-amino-2-butenyl)methylamino)-5′-deoxyadenosine (AbeAdo) and its 8-methyl derivative (Genz-644131) [263,265]. These inhibitors have not yet been proven to be effective antitumor medicines though they are promising for treating trypanosomiasis. Similarly, inhibitors of the next step (aminopropyl transferases) have been demonstrated to lower polyamine content [266], but effective inhibitors have not yet been identified for clinical applications.

The most well-known polyamine inhibitor, difluoromethylornithine (DFMO), was found in the 1970s and inhibited ODC irreversibly [267,268,269]. ODC is permanently rendered inactive once DFMO attaches to it, creating an extremely reactive intermediate that is then decarboxylated and covalently bound to ODC [269]. DFMO reactions result in polyamine depletion and are typically cytostatic in mammalian cells [270]. Its rapid clinical trial evaluation as a separate treatment agent was prompted by early observations of the impacts of DFMO in colon cancer, melanoma, small-cell lung cancer, and neuroblastoma [271,272,273,274,275]. Despite the fact that DFMO was well tolerated, the outcomes did not include notable clinical responses, which may have been the result of its ineffective distribution to cells [128,271]. Therefore, research centered on DFMO in combination with other medicinal drugs. In prostate, melanoma, breast, and neuroblastoma cell lines, the effective transport inhibitor AMXT 1501 synergizes with DFMO [276,277]. Patients with glioma have received DFMO in combination with the cytotoxic drugs procarbazine, nitrosourea, and vincristine, while those with neuroblastoma have received DFMO in combination with either bortezomib or etoposide, a proteasome inhibitor [278,279,280]. In vitro and in vivo, gemcitabine-resistant pancreatic cancer is efficiently inhibited by DFMO in conjunction with the polyamine transport inhibitor Trimer44NMe [126]. A promising method to treat colorectal cancer in an in vivo model has been demonstrated to be preventing ODC expression by DMFO as a separate agent or together with other medications, which is yet to be evaluated in clinical trials [281,282,283].

Additionally, the discovery of inhibitors has focused on the spermine and spermidine synthases S-adenosyl-1,12-diamino-3-thio-9-azadodecane (AdoDATAD) and S-adenosyl-3-thio-1,8-diamino-3-octane (AdoDATO) [284,285]. However, although these substances effectively and selectively block aminopropyl transferase, they both have primary amines in their structures that act as SSAT and amine oxidase substrates. Their clinical usage is therefore constrained because cellular metabolism breaks them down. These inhibitors only marginally reduce the growth of cancer cell lines [284,285]. An alternative spermine homolog, None-carbonyclopropyl-methyl-N11-ethylnorspermine (CPENSpm) [286], is significantly cytotoxic to breast cancer and human lung carcinoma cells [287,288,289]. It results in the induction of elevated SSAT levels and the activation of apoptosis [289,290]. However, no clinical trials have yet been completed, largely due to the drug’s poor cell type-specific cytotoxicity.

In experimental animal models, drugs, or polyamine analogs that target polyamines and important enzymes connected to polyamine metabolism have been found to be beneficial against cancer. Some of these drugs have also been tested in human clinical trials. However, as far as we can tell, these inhibitors’ adverse effects and toxicity have prevented them from producing adequate clinical results to date. Despite significant advancements in creative polyamine analogs and other polyamine-targeting drugs, the production of effective and secure therapeutic agents still needs further investigation.

## 6. Conclusions

Researchers’ interest in cancer metabolism has increased over the past decade, which has resulted in a greater understanding of the metabolic pathways involved in cancer biology. Numerous pathways that are known to or are anticipated to increase the survival of cancer cells rely on one-carbon and polyamine metabolism. A more thorough comprehension of these could enable more focused targeting of the particular pathways that are most crucial for cancer cell survival. There are already several therapies that target one-carbon and polyamine metabolism. However, due to the significance of one-carbon and polyamine metabolites in healthily proliferating cells, it has been challenging to avoid harmful side effects. Nonetheless, there are encouraging prospects for therapies that deplete serine and methionine, particularly in combination with redox or autophagy intervention. Altering methionine or SAM levels has significant effects on cancers, but currently, the mechanisms responsible are unclear, so further work is needed to develop specific and effective interventions. Polyamine-targeting drugs have been in clinical use for decades, and there are ongoing trials to optimize their use in combinations such as with NSAIDs in colorectal cancer. By more specifically blocking individual one-carbon and polyamines pathway enzymes, future treatments may be able to target one-carbon and polyamine metabolism more effectively in cancer cells. Therefore, this review strongly suggests the need for further investigations to explore a better understanding of one-carbon and polyamine metabolic pathways, particularly methionine and polyamine metabolism in cancer growth, and to discover novel inhibitors in these pathways.

## Figures and Tables

**Figure 1 biomolecules-12-01902-f001:**
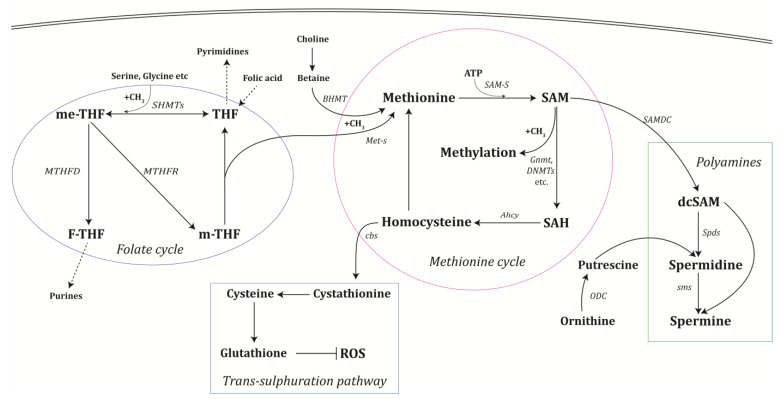
One-carbon metabolism includes the methionine cycle, which is linked to the folate cycle, polyamine synthesis, and the trans-sulphuration pathway. Enzymes catalyzing significant reactions are shown in italics. Metabolite abbreviations are: SAM: S-adenosyl methionine; SAH: s-adenosyl homocysteine; dcSAM: decarboxy-s-adenosyl methionine; ROS: reactive oxygen species; THF: tetrahydrofolate; m-THF: methyl-THF; me-THF: methylene-THF; F-THF: formyl-THF.

## Data Availability

Not applicable.

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
