# Peer review of "One-Carbon and Polyamine Metabolism as Cancer Therapy Targets"

_biomolecules, 2022, doi:10.3390/biom12121902_

Round 1

Reviewer 1 Report

This review article aimed to identify one-carbon and polyamine metabolism as cancer therapeutic targets. The incidence of cancer is increasing worldwide, and the current clinical treatment do not achieve an idea effects on some cancers, therefore, exploring the relationship between nutrient metabolites and tumors, looking for a new anticancer medication targeting  1C metabolism in the future are very important and interested. There are several comments on the present article as follows

Your mentioned that one carbon metabolism includes the methionine cycle, which is mainly linked to the folate cycle, and also related polyamine synthesis and the trans-sulphuration pathway, however, choline-betaine cycle, by participating in the methionine cycle, is also an important and irreplaceable pathway of methyl metabolism. I would prefer you to add the relevant content in context and figure 1

You described that the original source of the methyl groups is primary from folate. I don’t agree with that. As I know, Choline and betaine are direct methyl donors, which donate methyl group to folate. So folate is just a methyl carrier, only when it gets a methyl group from choline, betaine or other methyl donors and converts to, methyl tetrahydrofolate can it act as a methyl donor role. You should clarify this in your article.

The hierarchy of this article is very vague. I would prefer the articles to be structured as introduction, Metabolism, relationship between metabolites and cancer, the related mechanism, cancer therapeutic or prevention with metabolites, conclusion

Several minor comments

1.    This article has many headlines, but only one headlines is numbered, the others are not numbered, so I don't know which is the headline and which is the sub-headline. Please re-number.

2.    Please be consistent in the description of methyl donor. One -carbon or 1-C throughout the paper

Author Response

You mentioned that one carbon metabolism includes the methionine cycle, which is mainly linked to the folate cycle, and also related polyamine synthesis and the trans-sulphuration pathway, however, choline-betaine cycle, by participating in the methionine cycle, is also an important and irreplaceable pathway of methyl metabolism. I would prefer you to add the relevant content in context and figure 1

We agree that the role of choline and betaine in providing methyl groups is important, particularly in the liver. We have altered figure 1 to show this input. The role of betaine homocysteine methyltransferase was introduced in lines 50-63, and we have now specifically pointed out the contribution of choline and betaine in contributing methyl groups via the folate cycle (lines 73 and 149).

You described that the original source of the methyl groups is primary from folate. I don’t agree with that. As I know, Choline and betaine are direct methyl donors, which donate methyl group to folate. So folate is just a methyl carrier, only when it gets a methyl group from choline, betaine or other methyl donors and converts to, methyl tetrahydrofolate can it act as a methyl donor role. You should clarify this in your article.

We thank the reviewer for pointing out this error. We intended to indicate that methionine synthase provides the majority of methyl groups in most tissues, so they are being delivered via folate, however we were incorrect to say that folate was the original source – clearly the original source was methylated amino acids such as serine, choline etc. Accordingly, we have altered the text to make this clear.  We deleted line 96 and added a sentence regarding the original source of methyl group (lines 72 to 74). 

The hierarchy of this article is very vague. I would prefer the articles to be structured as introduction, Metabolism, relationship between metabolites and cancer, the related mechanism, cancer therapeutic or prevention with metabolites, conclusion

As suggested, we have changed the hierarchy of our manuscript. Please see the revised version which now follows the suggested structure.

Several minor comments

  1. This article has many headlines, but only one headlines is numbered, the others are not numbered, so I don't know which is the headline and which is the sub-headline. Please re-number.

As suggested, we have numbered the headings and subheadings.

  1. Please be consistent in the description of methyl donor. One -carbon or 1-C throughout the paper

As suggested, we have used one-carbon throughout the paper.

Reviewer 2 Report

Review: One-carbon and polyamine metabolism as cancer therapy targets

In this review manuscript by Islam et al., the authors seek to summarize the fields of both one-carbon and polyamine metabolism and how each represents unique vulnerabilities in tumors for targeting purposes.

This review has a few sections that are well described and cited appropriately, but several areas should be expanded and better organized. There is also a lack of citations to the original papers which described the targeted therapies and discoveries.

If the authors choose to review both one-carbon and polyamine metabolism, the organization of the manuscript should be considered by covering all aspects of one-carbon metabolism in a section before jumping to polyamine metabolism.

The conclusions are underdeveloped and should highlight a couple of significant discoveries in each respective field and be forward-thinking about the most promising therapeutic approaches.

In general, authors should choose to either cover one-carbon metabolism for cancer targeting and remove polyamine metabolism or the opposite. Unfortunately, the polyamine metabolism section is underdeveloped and lacks an appropriate introduction. Additionally, polyamine metabolism is not mentioned in the abstract, as the focus appears to be one-carbon metabolism.  

While this manuscript shows insight into reviewing one-carbon and polyamine metabolism, significant issues in appropriate coverage of polyamine metabolism/therapy, organization of the manuscript, and inclusion of critical references prevent an acceptance of the manuscript in its present form.

Author Response

This review has a few sections that are well described and cited appropriately, but several areas should be expanded and better organized. There is also a lack of citations to the original papers which described the targeted therapies and discoveries.

Following the reviewer’s suggestion, we have expanded and re-organized the manuscript. The organization follows the suggestions from reviewer 1 and expansions can be seen in several areas including lines 480-485 and 599-612, a total of 27 additional lines. We have added references to cite original research, including references 33, 35, 50, 51, 65, 238 and 239. We regret that it is not always possible to cite all the original work – we already have nearly 300 references.

If the authors choose to review both one-carbon and polyamine metabolism, the organization of the manuscript should be considered by covering all aspects of one-carbon metabolism in a section before jumping to polyamine metabolism.

Following to the reviewer’s suggestion, we have rearranged the metabolism section so that polyamine metabolism is not addressed until one-carbon metabolism is complete.

The conclusions are underdeveloped and should highlight a couple of significant discoveries in each respective field and be forward-thinking about the most promising therapeutic approaches.

Following the reviewer’s suggestion, we have expanded the conclusion section of our manuscript, including what we consider to be the best prospects for new therapies.

In general, authors should choose to either cover one-carbon metabolism for cancer targeting and remove polyamine metabolism or the opposite. Unfortunately, the polyamine metabolism section is underdeveloped and lacks an appropriate introduction. Additionally, polyamine metabolism is not mentioned in the abstract, as the focus appears to be one-carbon metabolism.  

We recognise that it is challenging to cover two areas in a single review – inevitably the coverage is incomplete, even with nearly 300 references. However, we feel that there are enough tightly focused reviews and not enough recognition of the link between one-carbon metabolism and polyamines. Following the reviewer’s suggestion, we have pointed out the relevance of polyamines in the abstract and clearly pointed out the polyamine introduction (section 2.3). We recognize that we have not discussed polyamine metabolism at length and have pointed the reader to good reviews on that subject (refs 16-20). Our focus has been on targeting polyamines for cancer therapy, which we cover in some detail (sections 3.7, 4.3 and 5.2)

While this manuscript shows insight into reviewing one-carbon and polyamine metabolism, significant issues in appropriate coverage of polyamine metabolism/therapy, organization of the manuscript, and inclusion of critical references prevent an acceptance of the manuscript in its present form.

We have significantly altered the organization of the manuscript according to the reviewers’ suggestions. We have added references to cite original research. We recognize that this is not a complete review on polyamine metabolism (we scarcely mention degradation), but our focus is on cancer therapy targets, not the underlying biology

Round 2

Reviewer 1 Report

Thanks for the point-by-point rebuttal letter and all the issues have been clarified.